# Theoretical and Experimental Analysis on the Influence of Rotor Non-Mechanical Errors of the Inductive Transducer in Active Magnetic Bearings

**DOI:** 10.3390/s18124376

**Published:** 2018-12-11

**Authors:** Jinpeng Yu, Yan Zhou, Ni Mo, Zhe Sun, Lei Zhao

**Affiliations:** 1Institute of Nuclear and New Energy Technology, Tsinghua University, Beijing 100084, China; yu-jp15@mails.tsinghua.edu.cn (J.Y.); zhou-yan@tsinghua.edu.cn (Y.Z.); moni@mail.tsinghua.edu.cn (N.M.); zhaolei@tsinghua.edu.cn (L.Z.); 2Collaborative Innovation Center of Advanced Nuclear Energy Technology, Tsinghua University, Beijing 100084, China; 3The Key Laboratory of Advanced Reactor Engineering and Safety, Tsinghua University, Beijing 100084, China

**Keywords:** active magnetic bearings, inductive transducer, rotor non-mechanical errors, anisotropic internal permeability, anisotropic surface conductivity

## Abstract

Inductive transducers are widely applied to active magnetic bearings (AMBs). However, when the rotor rotates at a high speed, the rotor defects will affect the measuring signal (the magnetic field generated by transducer coils) and then reduce the transducer measuring accuracy. The rotor in AMBs is assembled with laminations, which will result in rotor non-mechanical errors. In this paper, rotor non-mechanical errors, including the anisotropic internal permeability and anisotropic surface conductivity, and their influence on double-pole variable-gap inductive transducers are explored in depth. The anisotropic internal permeability will affect the transducer measuring accuracy and bring about 1.3±0.1% measurement error. The anisotropic surface conductivity leads to different eddy currents around the rotor, influences the equivalent reluctance of the magnetic circuit, and then affectsthe transducer measuring accuracy. The experiments prove that rotor non-mechanical errors have a significant influence on transducer measurement accuracy, and the reduction of the transducer excitation frequency can reduce the measurement error and improve the AMB control performance.

## 1. Intrduction

Active magnetic bearings (AMBs), which utilize magnetic levitation technology, are a type of bearing designed to support a load at a neutral position without mechanical contact [1,2]. AMBs possess several advantages, including an absence of friction, no need for lubrication, and a long-term high speed running ability, so they have the capability of active vibration control, allowing for the reduction of rotor vibrations [3]. Therefore, AMBs are widely applied in high-speed rotating machinery and flexible rotordynamic systems [4,5]. However, AMBs are open-loop unstable systems, so the rotor position must be accurately measured in real time to achieve closed-loop feedback control. Inductive transducers are widely applied to the AMB system with their characteristics of high precision and non-contact measurement [6]. Inductive transducers measure the moving target, which change the transducer inductance as a function of the target position. Therefore, the changing reluctance allows the transducer to be used as a length resolution [7].

In the measurement, if the target moves only along the measuring direction, as shown in Figure 1a, the measuring accuracy of inductive transducers will not be affected, because the measuring point remains unchanged. However, in the AMB system, the rotor both moves along the measuring direction and rotates at a high speed, as shown in Figure 1b. As for variable-gap inductive transducers, the measuring signal (the magnetic field generated by transducer coils) will pass through the surface and interior of the rotor. Therefore, in the AMB system, the measuring point of inductive transducers changes constantly and the measuring accuracy will be affected by the rotor defects.

There are many rotor defects that affect the measuring accuracy of inductive transducers, including the material inhomogeneity, impurities, and mechanical errors [8,9]. In the AMB system, the rotor is laminated with non-oriented silicon steel sheets, which are fixed by insulation materials, to reduce the eddy current, as shown in Figure 2. Although silicon steel sheets are non-oriented, the material permeability (in this paper, the permeability refers to the magnetic property in electromagnetism) still possesses difference at different directions. In addition, the laminations may have an insulation defect when assembling the rotor, and the lamination insulation on the rotor surface will be destroyed while lathing the rotor. All of this leads to the anisotropic conductivity of the rotor surface.

When the rotor rotates at a high speed, rotor non-mechanical errors, including the anisotropic internal permeability and anisotropic surface conductivity, will lead to physical properties of the measuring point changing continuously and then affect the measuring signal of inductive transducers. Therefore, rotor non-mechanical errors can affect the transducer measuring accuracy and lead to errors in displacement signals for AMBs. Inductive transducers have been studied in the aspects of structural designs [7,10], external environmental effects [11,12], and material properties of the target [8,9]. However, as for the variable-gap inductive transducers applied to AMBs, the influence of rotor non-mechanical errors on transducer measuring accuracy has not been studied in detail.

Rotor non-mechanical errors include the permeability, conductivity, piezoelectricity, etc. In this paper, the anisotropic internal permeability and anisotropic surface conductivity of the rotor and their influence on transducers and AMB controller are explored in depth. The transducer studied in this paper is a double-pole variable-gap inductive transducer, as shown in Figure 3a. In Section 2, the influence of anisotropic internal permeability on the transducer is studied with theoretical analysis and finite element analysis (FEA). Section 3 focuses on analyzing the anisotropic surface conductivity and its influence on the transducer. At last, the experiments are carried out to validate the influence of rotor non-mechanical errors on the transducer measurement accuracy and AMB controller.

## 2. Anisotropic Internal Permeability of the Rotor

In this section, the working principle of the variable-gap inductive transducer is introduced and the influence of the anisotropic internal permeability on the transducer is studied with theoretical analysis and FEA.

### 2.1. Model Establishment and Analysis

As shown in Figure 3b, the magnetic field generated by transducer coils will pass through the transducer core (stator), air gaps, and the target (rotor) to form a closed magnetic circuit. The reluctance of the magnetic circuit, the magnetic flux density, and the coil inductance will be affected by the air-gap length. Therefore, by measuring electrical parameters of the transducer circuit, the coil inductance can be measured, and the air-gap length can then be obtained. The transducer-rotor model can form a closed-loop magnetic circuit. Ignoring the magnetic flux leakage and magnetic hysteresis, the magnetic circuit can be calculated:(1)$$\Phi =\frac{NI}{\sum R_{i}}$$
with Φ as the magnetic flux, *N* as the total coil number, *I* as the coil current, and $\sum R_{i}$ as the total reluctance. In this paper, the coil is current driven, and the current frequency is 20KHz. The relationship between Φ and the coil inductance *L* is
(2)NΦ=LI.

Therefore, according to Equations (Equation 1) and (Equation 2), the coil inductance is
(3)$$L=\frac{N^{2}}{\sum R_{i}}.$$

It can be seen that the coil inductance is a function of the reluctance $\sum R_{i}$, which includes the reluctance of the rotor, stator, and air gaps:(4)$$\sum R_{i}=\sum \frac{l_{i}}{\mu_{i}S_{i}}=\frac{l_{1}}{\mu_{1}S_{1}}+\frac{l_{2}}{\mu_{2}S_{2}}+\frac{2\delta }{\mu_{0}S_{3}}.$$
$S_{i}\left(i=1,2,3\right)$ is the cross-sectional area of the magnetic circuit corresponding to the stator, the rotor, and air-gap length, respectively. $\mu_{1}$ and $\mu_{2}$ are the permeability of the stator and rotor, and $\mu_{0}$ is the air-gap permeability. $l_{1}$ and $l_{2}$ are the length of the magnetic circuit corresponding to the stator and the rotor, and δ is the air-gap length.

In order to reduce the computation, the magnetic circuit model can be simplified as shown in Figure 3b. The stator is located in the stationary coordinate system (defined as the XOY coordinate system), while the rotor is located in the rotating coordinate system (defined as the $X^{\prime }OY^{\prime }$ coordinate system). The parameters of the magnetic circuit in this section can be obtained according to the Figure 3b, as shown in Table 1.

In the experiment, the magnetic flux density in the magnetic circuit is less than 100mT, under which the material permeability in this paper is about $\mu_{s}=3000\;\mu_{0}$ [13]. In practical applications, the permeability possesses an error about ±10%. For the convenience of calculation and analysis, the rotor permeability is rewritten by Fourier decomposition:(5)$$\mu_{2}=\mu_{s}+\sum_{n=1}^{\infty }\left(\alpha_{n}\cos n\theta +\beta_{n}\sin n\theta \right),$$
where θ is the angle position of the rotor. Select the first-order term and set the maximum error as $\mu_{e}=300\;\mu_{0}$:
(6)$$\mu_{2}=\mu_{s}+\mu_{e}\sin\left(\theta +\varphi \right)$$
where φ is initial phase angle. Without loss of generality and based on the coordinate systems in Figure 3b, let the maximum rotor permeability $\mu_{2a}=\mu_{s}+\mu_{e}=3300\;\mu_{0}$ be in the X’ direction and the minimum $\mu_{2b}=\mu_{s}-\mu_{e}=2700\;\mu_{0}$ in the Y’ direction. The relationship between the permeability and the angle position of the rotor is shown in Figure 4. The rotor is in the X’–X direction when the X’ axis coincides with the X axis and the Y’–X direction when the Y’ axis coincides with the X axis. The stator permeability will not change in the measurement, so it is set as $\mu_{1}=3000\;\mu_{0}$. All the permeability parameters are shown in Table 1.

The reluctance and coil inductance are calculated as shown in Table 2. Considering the anisotropic internal permeability, the rotating rotor will lead to displacement error at the same air-gap length. The error can be up to 0.18±0.01%, which can affect the transducer measuring accuracy.

### 2.2. FEA for Anisotropic Internal Permeability

In order to verify the theoretical result, the 2D transducer-rotor model shown in Figure 3b is built in Ansys (licence number: 235XX) for FEA. Since the coil inductance cannot be measured in transient simulation, the air-gap magnetic flux density is used to substitute for the coil inductance.
(7)$$B_{gap}=\frac{NI}{S\sum R_{i}}.$$

According to Equations (Equation 1) and (Equation 7), the coil inductance *L* and the air-gap magnetic flux density *B* are both functions with respect to the circuit reluctance, so $B_{gap}$ can be used to reflect the change of coil inductance in the FEA. The current frequency is 20KHz, and the current amplitude is 0.5A. The FEA runs for $2.5\times 10^{-4}\;s$, and the data is selected in the last $5\times 10^{-5}\;s$ when the coil current is stable.

When the rotor is in the different directions, the average and maximum air-gap magnetic flux density are obtained in the FEA as shown in Table 3. It can be seen that the rotating rotor results in a difference of 1.3±0.1% for the mean and 7.8±0.1% for the maximum, which means that the anisotropic internal permeability does have an influence on the transducer measuring accuracy. However, it can be seen from the result that the error in the FEA is significantly larger than that in the theoretical calculation.

The different results mainly results from the cross-sectional area and the length of the magnetic circuit. Figure 5a shows the magnetic field distribution at the maximum coil current when the rotor is in the X’–X direction. It can be seen that the flux is mainly concentrated between poles and is not distributed evenly in depth, which is significantly different from the calculation. The effective cross-sectional area S is only about 1/10 of that in the theoretical analysis, and $l_{1}$ and $l_{2}$ are also affected.

Therefore, correct parameters of the magnetic circuit that $S_{re}=0.1S$, $l_{1re}=10\;\mathrm{mm}$, and $l_{2re}=30\;\mathrm{mm}$ and recalculate Equations (Equation 3) and (Equation 4). The coil inductions in the X’–X direction and the Y’–X direction are obtained as shown in Table 4. The difference is 0.9±0.1%, which is close to the simulation.

There are two main reasons for this phenomenon. Firstly, in the simulation, the coil current is 20KHz, so the high-frequency magnetic field generated by coils approaches the rotor surface. Secondly, the high-frequency magnetic field will result in an eddy current on the rotor surface, as shown in Figure 5b. The eddy-current magnetic field interacts with the transducer magnetic field and pulls the magnetic field close to the rotor surface [14,15].

## 3. Anisotropic Surface Conductivity of the Rotor

From the analysis above, in the measurement, the eddy current on the rotor surface will generate the magnetic field, which will influence the transducer measuring accuracy. Therefore, in consideration of the eddy current, this section mainly focuses on analyzing the anisotropic surface conductivity and its influence on the transducer.

### 3.1. Eddy-Current Magnetic Field Analysis

According to the Maxwell equations, the relationship between the eddy-current voltage *E* and the transducer magnetic flux density *B* on the rotor surface near the transducer poles is [16]
(8)$$E\left(t,x_{0},y_{0}\right)=\frac{d\;\int \int \;B(t,x,y)dxdy}{dt}=I_{eddy}\left(t,x_{0},y_{0}\right)R_{r}$$
where $I_{eddy}\left(t,x,y\right)$ is the eddy current in the rotor and $R_{r}$ is the rotor resistance. *B* can be resolved by Fourier decomposition:(9)$$B=\sum_{n=1}^{\infty }\left(\alpha_{n}\cos n\omega_{n}t+\beta_{n}\sin n\omega_{n}t\right).$$

The rotor resistance is
(10)$$R_{r}=\sigma \frac{l_{r}}{S_{r}}$$
where σ is the conductivity, and $l_{r}$ and $S_{r}$ are the length and cross-sectional area of the rotor resistance. Therefore, according to Equations (Equation 8)–(Equation 10), the eddy current $I_{eddy}$ is
(11)$$I_{eddy}=\frac{n\omega_{n}S_{r}}{\sigma l_{r}}\;\int \int \;\sum_{n=1}^{\infty }\left(\beta_{n}\cos n\omega_{n}t-\alpha_{n}\sin\omega_{n}t\right)dxdy.$$

The eddy-current magnetic flux density can be obtained according to Equations (Equation 11), i.e., the Boit–Savart–Laplace law [17,18]:(12)$$dB_{eddy}=\frac{\mu_{0}I_{unit}}{4\pi }\frac{d{\tilde{l}}_{\mathrm{unit}}\times {\tilde{D}}_{0\mathrm{unit}}}{D_{unit}^{2}}$$
where $I_{unit}$ is the unit-element current; d${\tilde{l}}_{unit}$—unit length of unit-element current; $D_{unit}$—distance between unit-element current and the checkpoint; ${\tilde{D}}_{0unit}$—unit vector from unit-element current to the checkpoint; $dB_{eddy}$—unit magnetic field that unit-element current generated on the watch point.

According to Equations (Equation 11) and (Equation 12), the eddy-current magnetic flux density is proportional to the eddy current, which is the function of conductivity σ. Therefore, with the fixed current frequency, the eddy-current magnetic field has a direct relationship with the rotor conductivity σ.

According to the Lenz Law, the eddy-current magnetic field aims at weakening the change of the transducer magnetic field, which will weaken the measuring signal. Therefore, the eddy current would have a side effect on increasing the reluctance of the magnetic circuit. In Equations (Equation 11) and (Equation 12), it can be seen that, under the same magnetic field and rotor parameters, the main factor affecting the eddy current on the rotor surface and its magnetic field is the surface rotor resistance. In practical applications, the surface rotor resistance only depends on the thickness of silicon steel sheets and the lamination insulation, which is the size of $l_{r}$. In the macroscopic sight, if the rotor surface is considered as an entire conductor, the direct influence of the lamination thickness and insulation is changing the surface conductivity of the rotor, and the insulation defect will lead to anisotropic surface conductivity. It should be noted, however, that this surface conductivity is not the σ in Equation (Equation 10) but is the equivalent conductivity for facilitating the analysis and calculation. The surface conductivity in this paper refers to this equivalent conductivity.

The anisotropic surface conductivity of the rotor will lead to an anisotropic eddy current, which will affect the transducer measuring accuracy. Therefore, the higher the eddy current, the more serious the influence is. There are several factors leading to anisotropic surface conductivity of the rotor. Firstly, the laminations may have an insulation defect when assembling the rotor, as shown in Figure 6a. The insulation defect can hardly be detected and eliminated. It can only be avoided by improving the assembling accuracy. Secondly, after assembling the rotor, it is necessary to improve the surface accuracy by lathing the rotor. The lamination insulation on the rotor surface will be destroyed when lathing the rotor, because the adjacent laminations will be pressed and conduct with each other on the rotor surface, as shown in Figure 6b. Some factors such as material properties and material impurities also have a certain influence on the homogeneity of the surface conductivity.

### 3.2. FEA for Anisotropic Surface Conductivity

The FEA is carried out to verify the influence of anisotropic surface conductivity on the transducer with the model in Figure 3b. In order to reduce the computation, the surface conductivity of the rotor affected by the lamination thickness and insulation defect are all equivalent to the material conductivity in FEA. Similarly, the average air-gap magnetic flux density is taken to reflect the coil inductance. The current frequency is 20kHz, and the other parameters are same as in Section 2.2. Similarly, the air-gap magnetic flux density *B* is used to reflect the change in coil inductance. The larger the eddy-current magnetic flux density is, the smaller the air-gap magnetic flux density is.

As shown in Figure 7, the simulation result shows that the eddy current on the rotor surface increases with the increase in conductivity under the same conditions. The ideal surface conductivity of the lamination stack is 0siemens/m.

As the solid line shown in Figure 8, with the constant current frequency, the average air-gap magnetic field reduces significantly with the surface conductivity increasing. The result proves that the anisotropic surface conductivity leads to different eddy currents around the rotor, influences the equivalent reluctance of the magnetic circuit, and then affects the transducer measuring accuracy.

### 3.3. Influence of Transducer Excitation Frequency

According to Equations (Equation 11) and (Equation 12), the transducer excitation frequency (coil current frequency) also affects the eddy current and its magnetic field. When the conductivity is fixed, the smaller the excitation frequency, the smaller the eddy current, and the smaller its influence on the transducer. The influence of the transducer excitation frequency on the measurement accuracy is simulated as shown in Figure 8. Similarly, under a fixed coil current, the average air-gap magnetic field reduces significantly with the increase in surface conductivity. When the conductivity is fixed, the average air-gap magnetic flux density decreases as excitation frequency increases. The higher the excitation frequency, the larger the eddy current and its magnetic flux density, which is consistent with theoretical analysis. Therefore, a lower transducer excitation frequency could reduce the influence of anisotropic surface conductivity on measurement accuracy.

## 4. Transducer-Rotor Experiment

According to the theoretical analysis and FEA in Section 2 and Section 3, the rotor non-mechanical errors, the anisotropic internal permeability, and the surface conductivity should influence transducer measuring accuracy. To further verify the influence, a transducer-rotor experiment was carried out. In the experiment, the double-pole variable-gap inductive transducer was studied, as shown in Figure 9.

### 4.1. Experimental Settings

In order to eliminate measurement errors caused by mechanical defects and the eccentric rotation of the rotor, the experiment was carried out directly on the lathe after lathing the rotor, as shown in Figure 10. After the rotor was lathed, the mechanical error was 0.01∼0.02 mm, which was measured by high-precision dial gauges before the experiment.
The rotor remained on the lathe after lathing, and the transducer probe was fixed on the bracket. The radial distance between the probe and the rotor (AMB part) was 1.00mm (measured by micrometer), and the probe was aligned with the center of laminations. The position was recorded as a zero point (recorded as 0.00mm).The probe was moved to the rotor by 0.40mm (record as −0.40mm) and away from the rotor by 0.40mm (record as 0.40mm), and the transducer output was recorded. The transducer sensitivity was calculated based on the data.The probe was moved back to 0.00mm, and the rotor was rotated to 80rpm. The transducer output was recorded, and the relationship between the transducer output (displacement) and the rotor rotation angle was obtained.

### 4.2. Original Results without Heating Treatment

According to the experimental data in Table 5, the transducer sensitivity is 0.110mm/V.

Figure 11 shows the relationship between the displacement and the rotor rotation angle. The peak-to-peak value of the output is 0.41mm. Since the mechanical error of the rotor is only 0.01∼0.02 mm, the transducer has a large measurement error of about 41±1%. Therefore, the rotor non-mechanical errors lead to a significant influence on transducer measurement accuracy.

### 4.3. Heating Treatment Experiment

To further prove that the measurement error is caused by rotor non-mechanical errors, the rotor surface was heated by acetylene.

Heating the rotor mainly has two functions. The insulation defect caused by rotor lathing can be solved by high-temperature heating on the rotor surface. When the temperature on the rotor surface is high, the contact edge of laminations can be reshaped and separated and the insulation will be improved, as shown in Figure 12. After the silicon steel sheets are heated, the internal grain distribution tends to be more disordered, which results in lower anisotropic internal permeability of the rotor.

Table 6 shows the transducer sensitivity after the rotor is heated. Transducer sensitivity after the first and the second surface heating treatment is 0.106mm/V and 0.104mm/V. It can be seen from the experimental data that the heating treatment has little influence on transducer sensitivity.

Figure 13a,b show the relationship between the displacement and the rotor rotation angle after the first and second heating treatment. The peak-to-peak value is reduced to 0.26mm and 0.19mm, respectively.

It can be seen from the experimental results that, in two-pole transducer measurement, the peak-to-peak values are significantly reduced after heating treatment. Heating treatment reduces the anisotropy of rotor surface conductivity and internal permeability and reduces the influence of rotor non-mechanical errors on the inductive transducer.

The experiment proves that rotor non-mechanical errors have a significant influence on transducer measuring accuracy, and the maximum error can be reduced by heating the rotor from 41±1% to 19±1%. However, in the experiment, heating the rotor surface did not eliminate transducer measurement errors completely. There are several reasons for this. Firstly, the assembling accuracy of laminations in the experiment is poor and leads to poor insulation inside the rotor, which needs to be improved in further experiments. Secondly, due to heating methods, temperature, heating time, etc., heating treatment cannot completely eliminate insulation defects on the rotor surface. Thirdly, since only the rotor surface is heated by acetylene, the internal temperature of the rotor is low, so the permeability anisotropy cannot be entirely reduced. To further reduce the rotor non-mechanical errors and its influence on the transducer, the lamination insulation must be improved significantly when assembling the rotor, and the heating method, such as the temperature control, should also be improved.

## 5. Solution for Rotor Non-Mechanical Errors

### 5.1. Frequency Influence on Transducer Measurement Accuracy

According to the theoretical analysis and FEA results, the influence of the anisotropic surface conductivity on the transducer measurement accuracy can be reduced by reducing the excitation frequency. Based on the transducer-rotor platform, the excitation frequency experiment was designed to determine solutions for rotor non-mechanical errors. The experimental device is the same as that described in Section 4.

The output of the transducer was measured at different excitation frequencies of 7.54KHz, 12.50KHz and 22.00KHz. The results are shown in Figure 14. When the current frequency was 22KHz, the transducer measurement error was about 35%. When the frequency dropped to 12.5KHz, the error clearly decreased to 13%. When the frequency further decreased to 7.5KHz, the error was 7%. The results show that the transducer measurement error can be reduced significantly by decreasing the transducer excitation frequency.

It should be noted, however, that, although reducing the transducer excitation frequency can reduce the measurement error, it does not eliminate the error. In addition, excessively decreasing the excitation frequency will also reduce the transducer resolution. Therefore, frequency reduction is an effective but not an ideal method to reduce the measurement error caused by rotor non-mechanical errors.

### 5.2. Improvement in the AMB Controller

In AMB-rotor systems, the transducer measurement error will be sent to the AMB controller, leading to error control in the rotor. Therefore, the rotor non-mechanical errors will eventually affect the dynamic performance of AMB-rotor systems. The AMB-rotor experiment was carried out to study the influence of frequency reduction on the AMB controller.

Figure 15 shows the axis orbit of the rotor at different transducer excitation frequencies of 22.00KHz, 12.50KHz, and 7.54KHz when the rotor speed is 8Hz. The red circle is the boundary of auxiliary bearings. It can be seen that at the same rotor speed, the rotor vibration drops significantly as the transducer excitation frequency decreases.
As shown in Figure 15a, when the transducer excitation frequency is 22.00KHz, the rotor vibrates severely at the speed of 8Hz, and the rotor cannot continue to speed up.After reducing the transducer excitation frequency to 12.50KHz, the rotor still has a large vibration, but the vibration is significantly less than that at 22.00KHz. The rotor can speed up to 40Hz, as shown in Figure 16a.When the transducer excitation frequency is 7.54KHz, the rotor vibration is significantly reduced. The rotor speed can easily rise to 50Hz and has a smaller axis orbit, as shown in Figure 16b.

The experiment further proves that the reduction in transducer excitation frequency can reduce measurement error caused by rotor non-mechanical errors and improve the performance of the AMB controller. However, frequency reduction is an effective, albeit not the only, way to solve rotor non-mechanical errors. Improving the lathing and assembling methods and selecting suitable material for the rotor are also important to eliminate the rotor non-mechanical errors at the source.

## 6. Conclusions

In this paper, the influence of rotor non-mechanical errors on the variable-gap inductive transducer is analyzed. The transducer-rotor model is established, based on which the influence of anisotropic internal permeability and anisotropic surface conductivity on the transducer measuring accuracy are analyzed and calculated.

Through theoretical calculation and FEA, the influence of anisotropic internal permeability on the transducer measuring accuracy is analyzed.
The measurement error is 0.9±0.1% in theoretical analysis and 1.3±0.1% in FEA, which means that the anisotropic internal permeability does have an influence on the transducer measuring accuracy.The result in Section 3 proves that the anisotropic surface conductivity leads to different eddy currents around the rotor, influences the equivalent reluctance of the magnetic circuit, and then affects the transducer measuring accuracy.The transducer-rotor experiment proves that rotor non-mechanical errors have a significant influence on transducer measuring accuracy, and the maximum error can be reduced by heating the rotor from 41±1% to 19±1% in a one-probe transducer.The transducer measurement error caused by rotor non-mechanical errors is reduced significantly by decreasing the transducer excitation frequency. However, the error can only be reduced but not eliminated.

Different rotors will have different materials and assembly methods. However, as long as the rotor is assembled by laminations, there will be insulation defects, inevitably leading to anisotropic surface conductivity. In addition, no matter what the rotor material is, there will always be anisotropic internal permeability caused by rolling treatment. Therefore, the study in this paper provides a new direction for transducers to improve the measuring accuracy, especially for transducers working with a high-frequency magnetic field.

However, this study only analyzes the influence of rotor non-mechanical errors from qualitative aspects. Further research is needed to separate rotor non-mechanical errors, the anisotropic internal permeability, and surface conductivity into different experiments so that they can be studied separately. More precise experiments are needed to measure the real material permeability and conductivity, so that theoretical analysis can be perfected. 

## Figures and Tables

**Figure 1 sensors-18-04376-f001:**
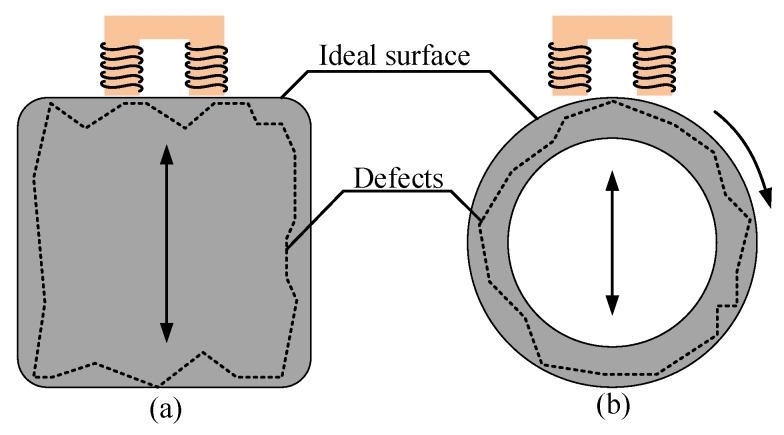
Different measuring situations. (**a**) The target moving along the measuring direction; (**b**) the rotor moving along the measuring direction and rotating at a high speed.

**Figure 2 sensors-18-04376-f002:**
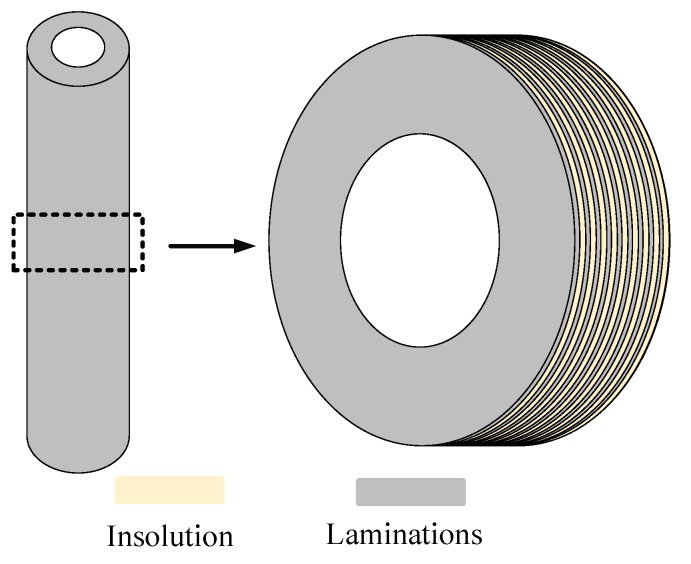
Rotor assembled with non-oriented silicon steel sheets and insulation materials.

**Figure 3 sensors-18-04376-f003:**
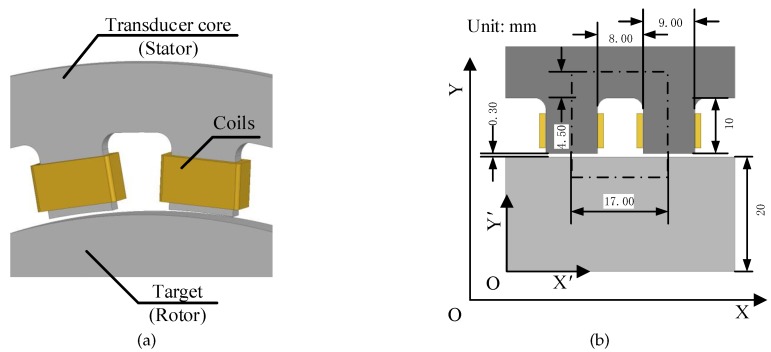
Double-pole variable-gap transducer-rotor model. (**a**) 3D model; (**b**) Simplified 2D model.

**Figure 4 sensors-18-04376-f004:**
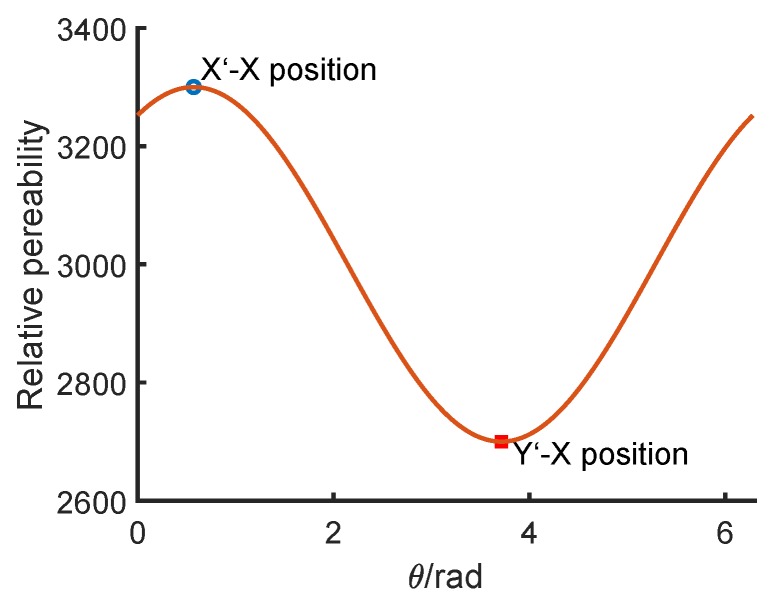
The simplified internal permeability distribution around the rotor.

**Figure 5 sensors-18-04376-f005:**
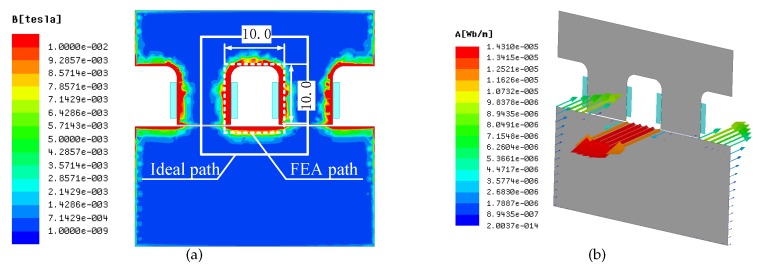
FEA results when the rotor is in the X’-X direction. (**a**) Magnetic flux density in the rotor and stator under high frequency coil current; (**b**) the eddy current on the rotor surface resulting from high frequency coil current.

**Figure 6 sensors-18-04376-f006:**
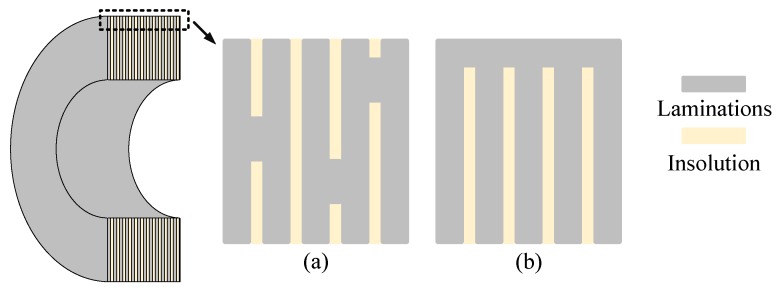
Lamination defects on the rotor surface. (**a**) Insulation defect caused by assembling the rotor; (**b**) lamination contact caused by lathing the rotor.

**Figure 7 sensors-18-04376-f007:**
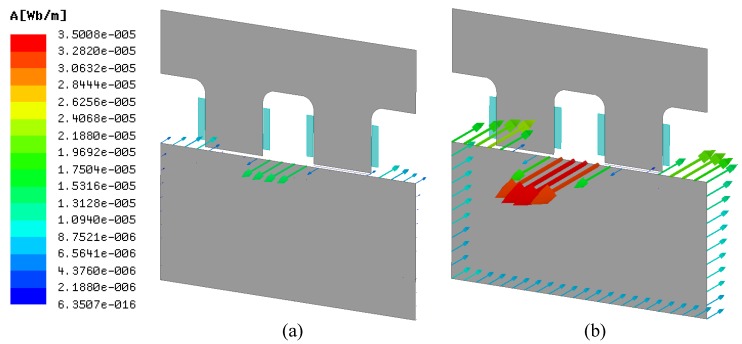
Variation in the eddy current on the rotor surface under different conductivity. (**a**) 2.5 × 10${}^{4}$ siemens/m; (**b**) 4.0 × 10${}^{6}$ siemens/m

**Figure 8 sensors-18-04376-f008:**
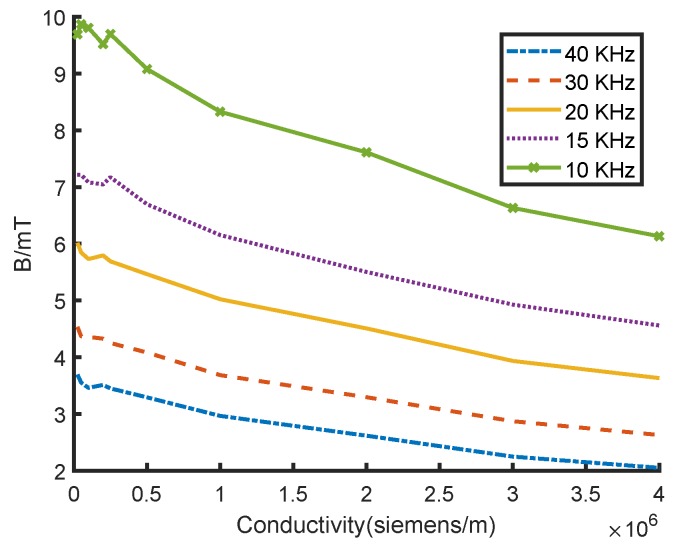
Variation in the average air-gap magnetic flux density under different surface conductivity and excitation frequency.

**Figure 9 sensors-18-04376-f009:**
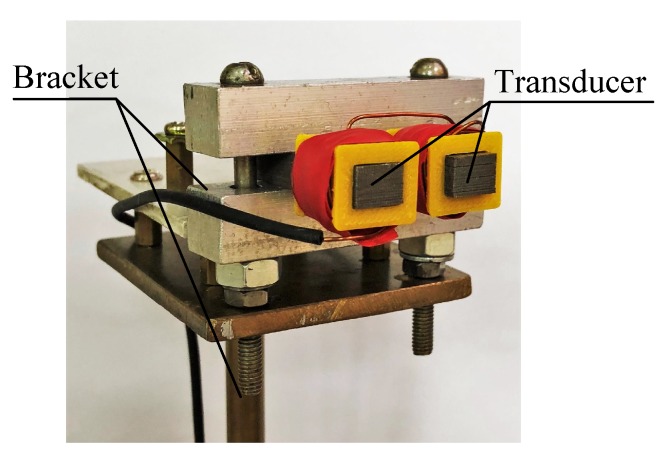
Double-pole variable-gap inductive transducer fixed on the bracket.

**Figure 10 sensors-18-04376-f010:**
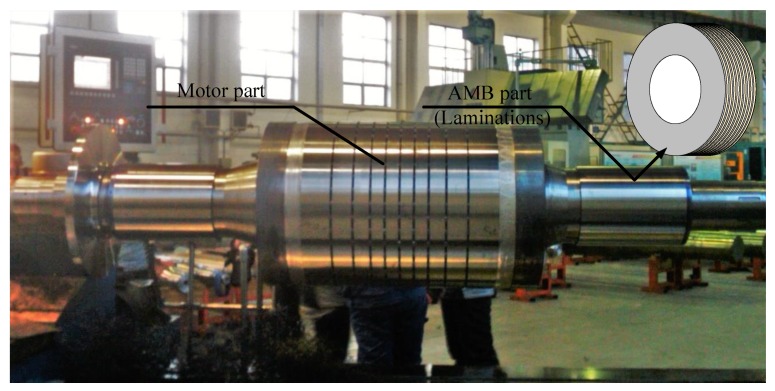
Laminated rotor with different functional parts in the experiment.

**Figure 11 sensors-18-04376-f011:**
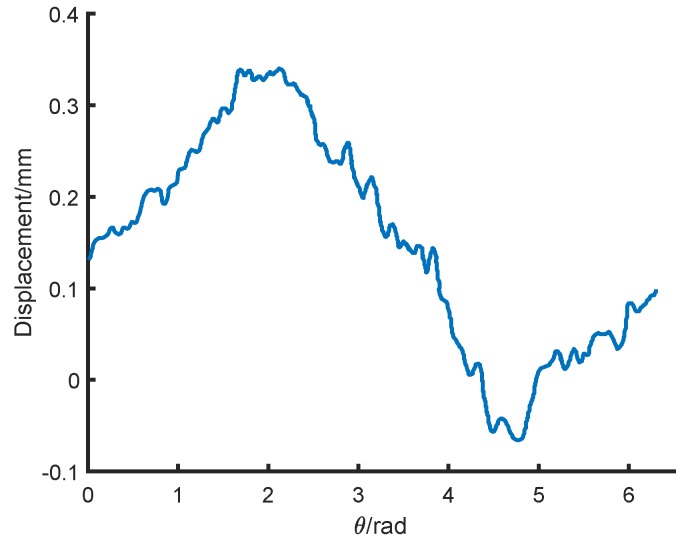
Relationship between the displacement and the rotor rotation angle under one-probe transducer measurement.

**Figure 12 sensors-18-04376-f012:**
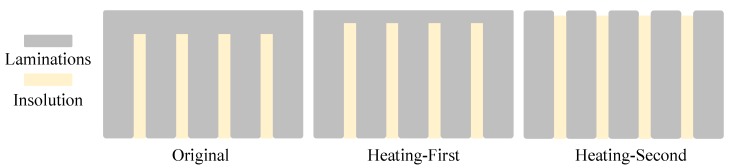
Decreasing rotor non-mechanical errors by heating treatment.

**Figure 13 sensors-18-04376-f013:**
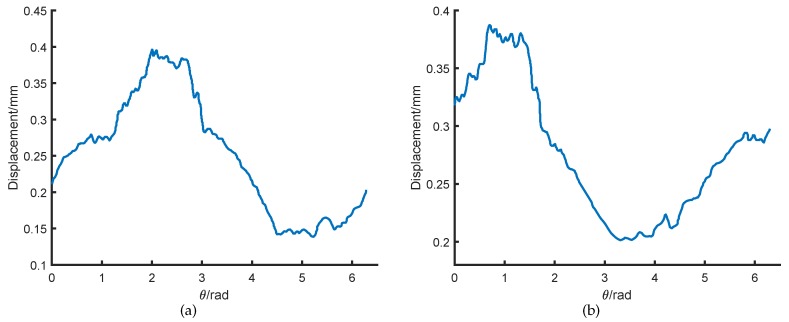
The relationship between the displacement and the rotor rotation angle in one-probe measurement. (**a**) The first heating treatment; (**b**) the second heating treatment.

**Figure 14 sensors-18-04376-f014:**
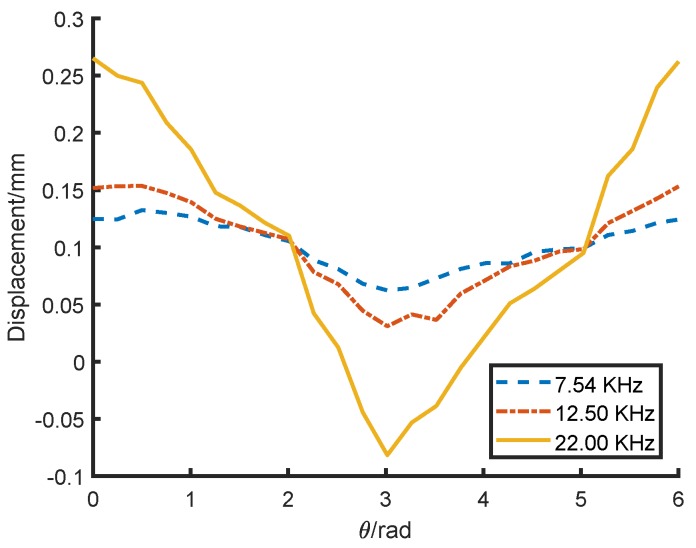
Reducing the measurement error by reducing the excitation frequency.

**Figure 15 sensors-18-04376-f015:**
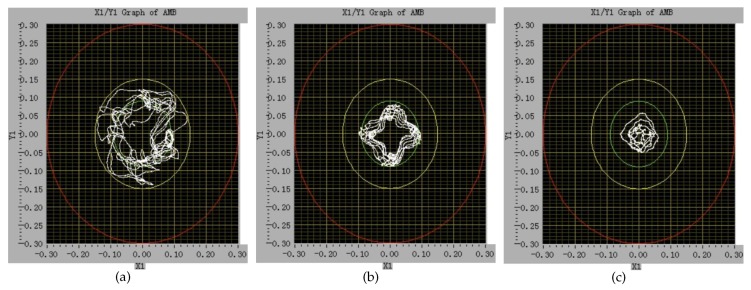
The rotor axis orbit at different transducer excitation frequencies when the rotor speed is 8Hz. (**a**) 22.00KHz; (**b**) 12.50KHz; (**c**) 7.54KHz.

**Figure 16 sensors-18-04376-f016:**
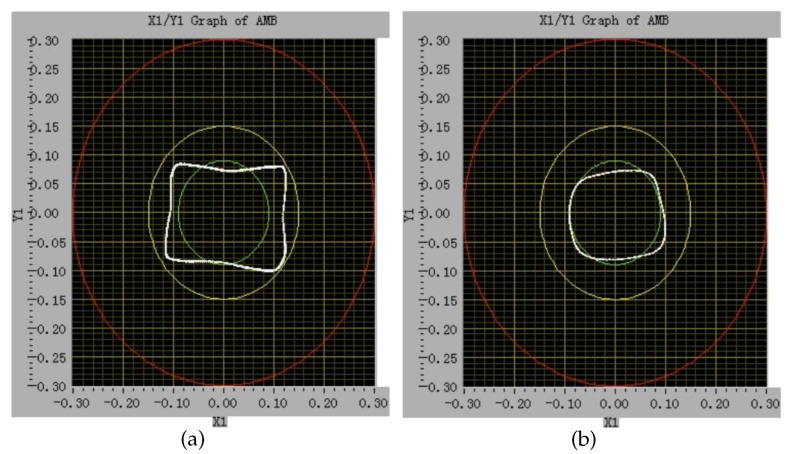
The rotor axis orbit at high rotor speed. (**a**) 12.50KHz transducer excitation frequency and 40Hz rotor speed; (**b**) 7.54KHz transducer excitation frequency and 50Hz rotor speed.

**Table 1 sensors-18-04376-t001:** The parameters of the magnetic circuit.

Parameter	Value	Unit	Parameter	Value	Unit
*N*	120		$S_{1},S_{2},S_{3}$	*S*
*I*	0.5	A	$\mu_{0}$	$4\pi \times 10^{-7}$	H/m
$l_{1}$	17.0	mm	$\mu_{1}$	3000	$\mu_{0}$
$l_{2}$	46.0	mm	$\mu_{2a}$	3300	$\mu_{0}$
δ	0.3	mm	$\mu_{2b}$	2700	$\mu_{0}$

**Table 2 sensors-18-04376-t002:** The magnetic circuit reluctance and coil induction at different rotor directions.

	X’–X	Y’–X	Difference
$R(H^{-1}m^{2}/S)$	494.017	494.928	0.18±0.01%
$L({\mathrm{Hm}}^{-2}\cdot S)$	7.287	7.274	

**Table 3 sensors-18-04376-t003:** Average and maximum air-gap magnetic flux density in the finite element analysis (FEA).

	X’–X	Y’–X	Difference
$B_{ave}/\mathrm{mT}$	5.107	5.043	1.3±0.1%
$B_{max}/\mathrm{mT}$	20.572	19.083	7.8±0.1%

**Table 4 sensors-18-04376-t004:** Air-gap errors in calculation and FEA.

	X’–X	Y’–X	Difference
FEA-$B_{ave}\left(\;\mathrm{mT}\right)$	5.107	5.041	1.3±0.1%
Calculation-$L(\;{\mathrm{Hm}}^{-2}\cdot S)$	6.134	6.192	0.9±0.1%

**Table 5 sensors-18-04376-t005:** Transducer sensitivity before heating treatment.

State	Displacement (mm)	Transducer Output (V)	Sensitivity (mm/V)
Original	0.00	0.23	0.110
	−0.40	3.84	
	0.40	−3.40	

**Table 6 sensors-18-04376-t006:** Transducer sensitivity after the first and second heating treatment.

State	Displacement (mm)	Transducer Output (V)	Sensitivity (mm/V)
First heating treatment	0.00	1.44	0.106
	−0.40	5.23	
	0.40	−2.35	
Second heating treatment	0.00	2.12	0.104
	−0.40	5.89	
	0.40	−1.83	

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
