# Peer review of "Theoretical and Experimental Analysis on the Influence of Rotor Non-Mechanical Errors of the Inductive Transducer in Active Magnetic Bearings"

_sensors, 2018, doi:10.3390/s18124376_

Round 1
Reviewer 1 Report
The work is about the measuring errors of inductive sensors for AMB applications. Although the topic is important for the field, my impression is that the investigation is still in very preliminary phase and can not be published at current stage.
Abstract:
What meaning to give 4 digit value in a 1% error evaluation?
Equations 1,2,3 there should be a clearer indication about the assumption at the base of these equations.
Figure 5. Not clear what is the meaning of X’-X position and Y’-X position indicated in the figure. The number of waves in one rotor rotation seems here to be 2. The reason of this assumption should be clarified. One could expect a multi-harmonic, although periodic distribution of the magnetic inhomogeneities and, consequently, something similar should occur in the reluctance.
Figure 6 and related comment. Sentence “magnetic induction near the rotor surface is significantly larger than that in the rotor” is not clear. It’s quite evident that flux is mainly concentrated between poles and is not distributed evenly in depth. Additionally:
- it is not clear why the magnetic flux density increases at the boundaries of the simulation domain, and
- what is the magnetic permeability distribution in the rotor that has been assumed in the magnetic model. One could expect a distribution depending from both X’ and Y’.
Statement: “Therefore, correct parameters of the magnetic circuit in the theoretical calculation, that Sre = 0.1S, l1re = 17 mm and l2re = 28 mm.” There should be an indication on how these values are found. This has also to do with a clear statement about the assumption about the lumped parameter magnetic model at the base of equations 1,2,3.
Statement “There are two main reasons for this phenomenon. Firstly, in the simulation, the coil current is 20 KHz, so the high-frequency magnetic field generated by coils approaches the rotor surface. ” There is no indication in previous part about the operating frequency of the coil and there is no indication wether if the coil is current or voltage driven. All this should be mentioned earlier because it is strongly affecting the lumped parameters model (equations 1,2,3).
Equation 5. Not Clear what the integration area is. Additionally, E is a voltage not an electric field.
Equation 6. What is dx*dy multiplying the harmonic terms?
Equation 7. What are S and l mentioned in the equation?
Apart from the qualitative aspects it is difficult to understand the quantitative use of equations 5-9.
Line 164 mentions that “the current frequency is 40kHz” but few lines earlier the mentioned frequency was 20kHz.
Figure 10: it could be useful to indicate what is the typical surface conductivity of lamination stack in the direction of the eddy current.
Figure 9: in previous text it is mentioned that lathing leads to an increased surface conductivity. Does the simulation take this into account? If yes, how?
Line 194: “Since the mechanical error of the rotor is only 0.01 ∼ 0.02 mm”, is there any reference probe to measure the mechanical inaccuracy?
Line 196: “In axial AMBs, the transducer has four probes distributing around the rotor evenly, as shown in Figure 13” It is not clear what “axial AMB” refers to. In fact Figure 13 looks like a probe arrangement typical of a radial bearing with x y measurement capability.
The experiments show some improvements in the sensor output after burning. Nevertheless, it is difficult from these tests to discriminate the contribution of the inhomogeneous magnetic properties from that due to the anisotropic conductivity. Although this is evidenced in the last sentence of the conclusions one could wonder what is the real contribution, apart from the experimental measurements and the effect of burning the surface. As a matter of fact the analytical and numerical models are not used to predict the errors that are then evidenced in the measurement.
Author Response
Thanks for your comments and suggestions. We have revised the paper carefully according to your suggestions and marked the changes with blue. The attachment is the revised file.

Reviewer 2 Report
Submitted manuscript is related to important technological area and it is in the score of the Sensors journal. It is a good attempt to estimate via comparative analysis of experimental and theoretical data on the inductive transducer errors origin in active magnetic bearings case. Despite the fact that subject is interesting and superficially the manuscript is very good illustrated it contains some deficiencies which must be corrected prior to possible publication of the work.
Below I propose some points to work with.
1. It is better to avoid any abbreviation in the title.
2. You must define “anisotropic internal permeability” clearly because it can be very different – from magnetic properties features to humidity.
3. How the value of the experimental error indicated in the abstract was calculated? “ I can not believe that it “bring about 1.269% measuring error”, it should be 1.3 ± 0.1 at most! Why other numbers for errors are much higher?
4. I would clearly define AMBs starting with most basic definition like “A magnetic bearing is a type of bearing that supports a load using magnetic levitation without physical contact...”
5. Referencing is very poor and must be extended by mentioning of significant works in the field (Meeks, C.R., "Magnetic Bearings - Optimum Design and Application", Paper presented at the International Workshop on Rare Earth Cobalt Permanent Magnets, University of Dayton, Dayton, Ohio, October 14–17, 1974; Basore P. A., "Passive Stabilization of Flywheel Magnetic Bearings," Master’s thesis, Massachusetts Institute of Technology (USA), 1980; R.Siva Srinivas, R.Tiwari, Ch.Kannababu, Application of active magnetic bearings in flexible rotordynamic systems – A state-of-the-art review, Mechanical Systems and Signal Processing Volume 106, June 2018, Pages 537-572; M.E.F. Kasarda An overview of active magnetic bearing technology and applications
Shock Vib. Dig., 32 (2) (2000), pp. 91-99; M.E. Johnson, L.P. Nascimento, M.E.F. Kasarda, C.R. Fuller The effect of actuator and sensor placement on the active control of rotor unbalance ASME J. Vib. Acoust., 125 (2003), pp. 365-373, O. Matsushita, T. Imashima, Y. Hisanaga, H. Okubo Aseismic vibration control of flexible rotors using active magnetic bearing ASME J. Vib. Acoust., 124 (2002), pp. 49-57, etc.).
6. All figure captions must be extended up to the level when they become to be understandable for readers – see Fig. 11 as an example completely unclear figure caption.
7. In all cases significan numbers must be corrected. For instance Table 4 contains displacements values like 0 and 0.4. It should be either 0 and 0 or 0.0 and 0.4.
8. I did not find clear indication about legal software used for calculations, the number of licence must be given in the text.
9. There are many misprints in the text and even misprints in the figures (see Fig. 10 as an example).
Author Response

(The authors gave the same response as above.)

Reviewer 3 Report
1. It is necessary to expand the experimental part;
2. Is it not clear whether this technique is universal? Each sample will have slightly different anisotropic conductivity.
Author Response

(The authors gave the same response as above.)

Round 2
Reviewer 1 Report
As most comments about previous version have been effectively addressed my suggestion is to publish as is.
Reviewer 2 Report
Work is better now and can be accepted in the present state. At the same time respected Authors must keep in mind that correct and sufficient referencing (not only their own works or contributions of a few groups) would certainly increase the interest of the other Authors.